# Exploring parents' experiences, attitudes and understanding of gastro-oesophageal reflux in infants

Kathryn McVicar[1], Lisa Szatkowski[1], Shalini Ojha[1,2], Simon Tunster[1], Manpreet Bains[1] *

1 Lifespan and Population Health, School of Medicine, University of Nottingham, Nottingham, United Kingdom, 2 Neonatal Unit, University Hospitals of Derby and Burton NHS Trust, Derby, United Kingdom

* manpreet.bains@nottingham.ac.uk

## Abstract

### Background

Gastro-oesophageal reflux (GOR) affects nearly half of infants. Parents play a crucial role in management but more understanding of their attitudes and experiences is needed to inform future education, support and research. This study aims to explore parental experiences, attitudes and understanding of the symptoms, diagnosis and management of infant GOR.

### Methods

Qualitative semi-structured interviews with 9 parents of infants with GOR in the UK, analysed by thematic analysis.

### Results

8 participants were mothers and median age was 34 years. Over half identified as White ethnicity. Parents described that GOR can affect all aspects of life, including mental well-being and bonding with their baby. Medications are time-consuming to prepare and can cause challenging side effects such as constipation. It is crucial that health professionals manage parental expectations in that treatments are not curative and symptoms do not last forever. Attitudes about healthcare professionals varied: some were perceived as dismissive, whilst some showed understanding. There were differences depending on whether the child was a first or second born child, with more understanding shown where the child was not the parents' first born. Parents felt more education could be beneficial for parents and clinicians.

### Conclusions

Infant GOR can affect infants and parents in a variety of ways, impacting both physical and mental health. Parents play a vital role in the management of infant reflux, but there is lack of consistency of information and levels of knowledge among healthcare professionals vary.

**Data Availability Statement:** Data underlying the findings is freely available via a public repository, accessible here https://osf.io/zm86g/.

**Funding:** The author(s) received no specific funding for this work.

**Competing interests:** The authors have declared that no competing interests exist.

More education could be beneficial, and further research is needed into health professionals' perceptions and fathers' experiences.

## Introduction

Gastro-oesophageal reflux (GOR) is defined as 'the involuntary retrograde passage of gastric contents into the oesophagus with or without regurgitation' [1]. It is thought to affect just under half of infants (children aged under one year) at least once a day because the lower oesophageal sphincter (LES) has not yet matured [1–3]. Reflux is especially common in preterm infants (born ≤37 weeks' gestation) but affects infants born both preterm and at term. Symptoms usually start before a child is eight weeks old and normally improve with age [2]. Whilst infants do not experience severe complications and continue to thrive, GOR is considered physiological and is usually self-limiting [2, 4]. However, reflux can occur with increasing frequency and severity which can damage the oesophageal mucosa and can cause pathological (or 'troublesome') symptoms or complications [1, 2, 4]. This is known as gastro-oesophageal reflux disease (GORD) [2]. GOR and GORD are often seen on a continuum, and it is often difficult to distinguish between them given the unclear boundary between physiological and pathological symptoms [1, 5].

Diagnosis of GOR or GORD in infants is most commonly based solely on the clinical history provided by the parent or carer [3]. The most common sign is frequent spitting up or regurgitation after feeds, but signs can be non-specific and often overlap with 'normal' infant behaviour, such as excessive crying or distress [3, 5]. This can make reflux difficult to diagnose and manage, and the condition has been described as 'commonly misunderstood' [6]. Crying infants are often overlabeled as having the condition and overprescribing of medications may be occuring [7–9]. The use of the label 'GORD' itself can sometimes affect parental decisions and expectations regarding pharmacological treatment [10].

There is a lack of clear, evidence-based guidelines for diagnosis and management of GOR/GORD [9, 11, 12], potentially due to the lack of gold standard objective measures and reliance on the reporting of signs and symptoms [1]. The National Institute of Health and Care Excellence (NICE) guidance for the management of GOR/GORD in infants recommends parental reassurance and education on conservative management, such as keeping an infant upright after feeds and offering smaller, more frequent feeds as first-line options [2]. A number of studies have shown the importance of parents in both diagnosis and management [5, 6, 13], but research into parental knowledge and experiences of GOR/GORD is limited (e.g. in the United Kingdom).

Having an infant with GOR/GORD has been shown to be distressing to parents and can affect parental wellbeing, leading to high levels of anxiety [14]. Infant irritability is a common symptom of GOR/GORD and this has been linked to maternal fatigue, anxiety and depression [15]. Additionally it could negatively impact bonding between the infant and parent during feeding, with mothers of infants with GOR/GORD experiencing worse interactions with their child than those without [5, 16].

Much of the research into infant GOR/GORD and the role of parents in the diagnosis and management has not been conducted in the United Kingdom, but other nations have identified that a greater understanding of parents' knowledge of, and attitudes towards the condition is needed to further understand the pressures on parents [5, 6, 9]. Given the recognised importance of parents in the diagnosis and management of GOR/GORD, and lack of exploration

into their attitudes and experiences, further research is needed [5, 6, 13], to understand whether findings from other settings are transferable to the UK, and to identify where gaps in knowledge are and ways to better support parents in the diagnosis and management of GOR/GORD (including who could input into this and how). The purpose of this study is to explore parental experiences, attitudes and understanding of the symptoms, diagnosis and management of infant GOR and GORD. This study is needed to begin to identify ways forward for future approaches to parental education and support, which would faciliate GOR/GORD management [17].

## Methods

This was a qualitative study involving semi-structured interviews with parents in the United Kingdom of children with GOR/GORD. A constructionist approach was taken, which assumes that meaning and experience are socially constructed [18, 19]. This allowed us to gain understanding of participants' experiences, recognising that findings would be co-constructed by the researchers and participants [20].

Ethical approval for the study was granted by the University of Nottingham Master of Public Health Ethics Committee on 28th September 2022.

### Study population, sampling and recruitment

A combination of convenience (self-selecting) and purposive sampling approaches were used to recruit a range of parents including those with infants born preterm and at term. Participants were recruited between 01 December 2022 and 17 May 2023 via adverts posted on the online forums of Mumsnet and Facebook, a poster at a children's playgroup in Nottinghamshire, and through adverts circulated through local parents' WhatsApp groups. Participants were eligible if they lived in the UK, were aged 18 or older, able to complete the interview in English, and were a parent or guardian of a child aged under 2 years at the time of interview who was diagnosed with GOR/GORD before their first birthday.

### Interview guide and procedure

The semi-structured interview guide (see S1 File) was developed based on the research aims, existing literature and NICE guidance [2]. The main topics covered were knowledge and understanding of infant GOR/GORD, diagnosis experiences, and experiences and attitudes about the management of GOR/GORD.

Prior to the interview, participants were sent a Participant Information Sheet, given the opportunity to ask questions, and completed a consent form via Microsoft Forms.

Interviews were completed by telephone or online (primarily Microsoft Teams) depending on participant preference and were recorded. A short demographic questionnaire was completed at the start of the interview. Participants received a £15 voucher for participation. The first interview served as a pilot resulting in amendment of one question to improve clarity.

### Data analysis

Interviews were transcribed verbatim by KM and transcripts were checked to ensure accuracy and to anonymise any personally identifiable data. Each participant was assigned a unique code which was ascribed to the corresponding transcript and audio file.

Data was analysed thematically using an inductive approach which allowed meaning to be generated from the data [21, 22]. Firstly, transcripts were read and re-read to develop

familiarity, and initial reflections were documented. Next, NVivo [23] was used to facilitate axial and analytical coding [24]. These initial stages facilitated interpretations at the semantic level [25].

Themes were then formed by collating codes with similar characteristics. At this stage analysis moved to a latent level [21]. Thematic maps were developed using NVivo to organise themes and corresponding subthemes. A small number of transcripts were independently double-coded by MB and ST to enhance validity [26]. Themes were reviewed using an iterative approach, where they were adapted until they accurately represented the data.

## Results

### Participants

Between December 2022 and May 2023, twelve individuals expressed interest in participating; nine were interviewed and three were unreachable. Interviews averaged 32 minutes (ranging between 18 and 46 minutes).

The median age of participants was 34 years (interquartile range, IQR, 30–36 years) and the median age of their child was 8 months (IQR 4–17 months). All but one of the participants were the child's mother and one was the child's father. Five were White British, three identified as White Other and one was Black British. All nine participants were educated to at least A-level equivalent, seven to at least Bachelor's degree level and four to Master's degree level. Five participants had one child; the remainder had two. Five of the participants' infants were born at term and four preterm.

All but one of the participants' infants were diagnosed with GOR/GORD within the first eight weeks of life (median 4 weeks, IQR 2–7). Symptoms commonly described by parents were regurgitation of milk, discomfort, crying, breathing difficulties and difficulty gaining or maintaining weight. All participants understood that reflux occurs due to passage of gastric contents into the oesophagus, and 5 also demonstrated knowledge and/or experience of silent reflux, where regurgitation of milk is not visible. All had tried conservative (or home-management) methods, such as smaller more frequent feeds, as well as some form of medical management for their infant's reflux; for two this was milk thickener only, seven had tried Gaviscon and six had also tried omeprazole.

### Overview of themes

Four themes with corresponding sub-themes were identified (Table 1). Each theme is outlined below with supporting quotes labelled with the participant identifier. No further identifying details are given to preserve anonymity.

## The impact of GOR/GORD on all aspects of family life

### Impact on activities of daily living, mental health and the role of support

Many parents described the challenges that having an infant with reflux adds to day-to-day life, such as leaving the house, seeing friends and family, and going to the toilet. Several parents described that they were unable to lay the child down as this was often when reflux symptoms were worst, resulting in them being unable to manage other tasks during the day, or affecting sleep at night, which caused stress.

*"It was very stressful because, you know, sometimes you might just want, 'oh, I just need to go to the bathroom' and especially when my husband went back to work and, like, I'll just put*

**Table 1. Themes and sub-themes.**

| Themes | Subthemes |
|---|---|
| The impact of GOR/GORD on all aspects of family life | • Impact on activities of daily living, mental health and the role of support<br>• Reality of parenting not matching expectations |
| The challenges in the management of GOR/GORD in infants | • Struggles with different medications<br>• Importance of managing expectations |
| The role of healthcare professionals in the diagnosis and management of GOR/GORD | • Importance of the attitudes of health professionals<br>• Overlap of healthcare professional role with parental role<br>• Lack of continuity of care |
| The need for more information and education on the symptoms and management of GOR/GORD | • For clinicians<br>• For parents and parents-to-be |

*you down and you know, he starts screaming because he's so uncomfortable, and it's like 'oh God', you know, just things like that. And so I did learn how to do a lot of things one handed."*

*(#5)*

Some parents described the toll that having an infant with reflux can have on mental health and wellbeing. Feelings of helplessness and self-blame were noted, and a few parents were emotional when discussing this, indicating the extent to which this experience affected them.

*"It would be very easy for your mental health to go really like really downhill, you know, you just had a baby, you're not going anywhere, you're not enjoying looking after them because they're miserable."*

*(#1)*

Parents mentioned the importance of having support, for example spousal support, and one parent indicated that a coffee morning with other parents who are going through the same experience 'would help mental health with parents astronomically.

*"So like supporting each other is very essential for mental well-being. I say this because on my own, I think if not for my [partner's] strength, I think the illness would have tortured me mentally a bit, but my [partner] was really strong."*

*(#6)*

## Reality of parenting not matching expectations

Parents explained the impact that infant GOR/GORD had had on their parenting experiences, and how consequently this had not matched up to their expectations, as parents explained that the symptoms of reflux had affected their bonding with their child. They felt their lived experiences, such as baby 'crying' or 'screaming' had adversely affected their ability to bond with their child, which mothers reported was more profound for their partners.

*"I suppose the other notable thing is that my husband has found it really difficult. . . there was a period of time where [child's name] just cried or actually screamed every time [father's name] held him. . . And [father's name] found it quite difficult to bond with him."*

*(#2)*

Many mothers spoke about their breastfeeding experiences, and the particular challenge of having to express milk to be able to administer medications such as Gaviscon and omeprazole (which need to be mixed with milk).

*"It was harder because we had to express the milk and put the Gaviscon powders in and then measure everything together and he was still spitting it out."*

*(#9)*

Some of these negative experiences impacted decisions about feeding for subsequent children.

*"One of the reasons I haven't breast fed the second time round is because it was not a pleasant experience all round. . . things like giving him the Gaviscon meant that you either give a bit of formula before the feed or express the milk with the Gaviscon before then feeding him, so it was essentially having to breastfeed him and have a bottle. . . it was a real nightmare."*

*(#1)*

## The challenges in the management of GOR/GORD in infants

### Struggles with different medications

All parents experienced similar challenges with medical treatments for GOR/GORD, such as the time-consuming nature of preparation and administration. While some parents felt that Gaviscon improved symptoms, they were met with constipation, which added another layer of complexity. However, others felt it made their child worse by seeming to cause more discomfort and distress.

*"We were kind of going back and forth with [Gaviscon], cause we were trying to stop him sicking up, but then we were struggling because he couldn't push everything through. . . he was getting extremely constipated and he would be waking up screaming."*

*(#5)*

Six parents were eventually prescribed omeprazole after trialling Gaviscon, with most describing this leading to the biggest improvement in symptoms. However, some had challenges with accessing the different formulations prescribed, when paediatricians had prescribed preparations that are not routine for infants and pharmacies were reluctant to dispense.

*"The omeprazole really was a game changer . . . [but] I've had a couple of occasions where the omeprazole capsule has been prescribed and the pharmacy has refused to give it to me because they've said you can't give this capsule to a one year old baby. And I've said 'this is how I'm gonna do it and this is what the paediatrician has told to do' but I've had a pharmacy twice*

*refused to give me the actual prescription because they don't feel comfortable giving it to a baby, despite the fact that the paediatrician had prescribed it for her. So that's been frustrating."*

*(#8)*

## Importance of managing expectations

Most parents explained that, given the challenges in managing infant GOR/GORD, managing expectations is vital. Many described the importance of understanding the treatment options are not necessarily a cure and knowing that GOR/GORD does not last forever.

*"The most helpful thing for me and for us probably is the consultant being really honest about kind of managing our expectations. . . he was just quite honest and said, you know "he's a reflux-y baby, he will grow out of it, but the next few months are going to be really tough until he does."*

*(#2)*

## The role of healthcare professionals in the diagnosis and management of GOR/GORD

### Importance of the attitudes of health professionals

Parents described that they encountered different attitudes from health professionals towards GOR/GORD, for example some felt these were patronising and dismissive, which they perceived led to delays in diagnosis or inappropriate management. Conversely, some parents felt they were shown understanding, in particular from health visitors. Health professionals' practices also seemed to be influenced by their own beliefs towards medications for reflux, with some eager to prescribe immediately and some much more reluctant; sometimes this was also felt to be influenced by length of appointment, with medications seen as a 'quick fix'.

*"For my first when I went to see the doctors talk about it, it was very much like "oh is this your first, you know the babies do cry you know, they do get a bit unhappy and they do have a bit of a bit of sick", so you know, it was patronising. . . the doctors [didn't] believe me. . . it must have been between five and six months when we actually got some medication for it."*

*(#1)*

*"It was kind of overlooked by a lot of the health professionals at the beginning, just kind of calling it as 'babies will be babies, you know, babies cry, babies scream, babies arch their backs', so he was overlooked at the beginning by a lot of the health professionals."*

*(#3)*

*"With Gaviscon we were told in the hospital that they didn't want to medicate him. Gaviscon they said had bad side effects in babies."*

*(#5)*

From parents of two children who both experienced GOR/GORD, strong differences in attitudes were noted between their first and second child, with more understanding shown towards the second child.

*"With my daughter, though, because of the history with my son, they just listened to me right away."*

*(#3)*

## Overlap of healthcare professional role with parental role

Parents explained that they felt most care was driven by them, rather than by health professionals and that this seemed to be because they felt they had become 'expert' in the condition. Reading about the symptoms and their experiences online, and through talking to others, aided parents to lead conversations with clinicians and push for a diagnosis.

*"Just independent research I think, because like my GP's are lovely, they've been so great, they've been so helpful. But I wasn't satisfied with just 'OK, here, put them on some medication', you know, because medication helps the symptoms and it helps to relieve the like problem at the time, but long term, what's it going to do? So that's what prompted me to kind of look into reflux more as a whole."*

*(#3)*

## Lack of continuity of care

Specifically for preterm infants where multiple health professionals were involved in the child's care, parents felt that there was at times a lack of continuity and unclear roles and responsibilities for the management of GOR/GORD. This led to parents feeling that they did not have support available to them when they needed it.

*"I have found the GP really frustrating because they are always like 'it's a neonatal baby, I don't want to get involved, you need to see the paediatrician', but then I won't help you get an appointment to see the paediatrician. . .My health visitor wasn't great, I have to be honest. I think she kind of, again, was a bit like. . . 'well, you're under so many other people that you kind of like don't really need me', but there were times where I actually really did need some help and support, but well, there was no one giving it to me."*

*(#8)*

## The need for more information and education on the symptoms and management of GOR/GORD

### For clinicians

Most parents felt that there was a need for health professionals to have more training on infant GOR/GORD, particularly regarding distinctions between different diagnoses in infancy (such as cow's milk protein allergy, CMPA) and also related to evidence-based management. Some referred to the need for a clear definition or diagnostic criteria for infant reflux.

*"I feel as though there needs to be kind of more education on reflux, on underlying causes of it, on how to treat it, than just what's already available to our medical professionals. I don't know if you know, maybe some further like CPD or training could go and work wonders, and you know more research could work wonders. Because it did feel like, you know, I had to kind*

*of go off my own back and, you know, take things into my own hands to try and get to the bottom of what was causing the reflux in both of my children."*

*(#3)*

*"It maybe would be helpful to have a better definition of GOR. Then it would be helpful, maybe for healthcare professionals as well as for parents to say like this is a normal amount of regurgitation and this isn't and kind of needs attention or is troublesome."*

*(#2)*

### For parents and parents-to-be

Participants described that, in general, infant reflux is often misunderstood, particularly in relation to the amount of distress it can cause and the ways it impacts those affected, and therefore there could be utility in educating all parents-to-be.

*"Perhaps sensitising people on reflux, like even before they get children. Like maybe like maybe putting posters in hospitals, mostly in maternity wards or the maternity areas and yeah maybe using newspapers and social media to sensitise people because I realised that I wasn't the only one who didn't know about it."*

*(#6)*

There was also a need for education about GOR/GORD for parents with an affected infant, not only to help with management but also with managing expectations of symptoms and duration. Parents explained that sometimes knowledge was assumed (particularly by General Practitioners and health visitors), and that it would be useful to have more education on the different ways GOR/GORD can present, for example without visible regurgitation.

It was explained that it would be useful to have a reliable source of information, given that there is often mixed and conflicting information or advice. This is both on the internet through lack of credible online sources (or difficulty identifying sources which are credible), and from healthcare professionals (e.g., some suggesting to eliminate dairy, but others stating there is no evidence behind this).

Four parents had professional backgrounds which helped inform their understanding of reflux, but they recognised reliance on this is not an option for all parents. One explained that their professional training did not reflect their personal experience.

*"It would have been nice to have a bit more information. It was kind of just like he has reflux, there you go, bye. Whereas it would have been nice if there was like, I don't know, just like an information sheet or something. So you have one source of information, you don't need to like go Googling and try and find everything out, they just say 'here's what you do for reflux' or 'these are your options' . . . My understanding was a little bit different in terms of my training versus in real life experiencing it. It's been a lot different than I kind of was told that it is."*

*(#7)*

## Discussion

Study participants demonstrated good knowledge and understanding of the mechanism by which GOR/GORD occurs and its common symptoms, such as regurgitation and discomfort. There were differences between first-time parents and those with more than one child, indicating that knowledge and experiences obtained from having a child with GOR/GORD leads to increased trust from health professionals or perhaps the ability to frame the situation in a way that better aids support.

Infant GOR/GORD can affect all aspects of life for the family, which can have a significant impact on parental wellbeing. Difficulties with preparation and administration of medications, which are often perceived as ineffective, adds to the burden, particularly amongst breastfeeding mothers. Care was felt to be primarily driven by parents, and parents described that they took on a similar role to health professionals by becoming an expert in GOR/GORD through independent research.

The study indicates how extensively infant GOR/GORD can affect not only the symptomatic child but also the parents. Parents are often unable to carry out their normal activities, which can affect parental mental health and wellbeing. Previous studies have aligned with these findings, demonstrating that infant crying, sleep disturbance and feeding difficulties are associated with higher levels of parenting stress, maternal anxiety and depression [27–29]. However, most prior research focuses on the impact on the mother, whereas in this study mothers and the one father who participated indicated that reflux can also affect the mental wellbeing of fathers.

It is well recognised that feeding is an important bonding experience between parents and babies [30, 31]. Breastfeeding in particular is acknowledged as a vehicle to facilitate bonding [31, 32]. A small pilot study found that parental-infant interactions were significantly worse where infants had GORD than when they did not [16]. This study echoes these findings, with particular reference to challenges administering medications impacting feeding. A novel finding from our study is that GOR/GORD can impact a mother's decision to breast feed future children, an additional factor that could affect parental mental wellbeing [33].

The majority of participants had trialled some form of medical management for infant GOR/GORD, in contrast to published literature and guidance which state for most cases reassurance alone can suffice. This perhaps reflects prior findings that overprescribing might be occurring, although the self-selective nature of recruitment could have meant that participants' children were more likely to have GORD than GOR, and therefore warranted medical treatment [1, 4, 9, 34, 35]. Research has shown that once infants are labelled with having a 'disease' such as GORD, parents are more interested in using medication [10]. The parent-driven nature of reflux management could have therefore played a role as to why most participants had experience with medical treatments.

Participants in this study had mixed experiences with medical management methods. A recent Cochrane review concluded that there is very low certainty evidence regarding whether medications provide benefit in infants where conservative reflux management has proved ineffective [36]. This highlights the importance of ensuring parental expectations are managed. Parents explained the importance health professionals explaining that medications are not curative and providing reassurance that symptoms will not last forever.

Crucially, our work suggests a lack of consistency in the knowledge and attitudes among healthcare professionals regarding infant GOR/GORD. This could imply that it is not well-taught during training, or could echo findings which emphasise the need for clearer guidelines for diagnosis and management [11, 12]. Another novel finding from our work was the marked difference in attitudes from professionals between first-born and subsequent children which

has not been previously identified in this context. This could reflect the increased parental knowledge and experience of having a previous child with GOR/GORD, and links with the parent-driven nature of management. Parents who have experienced GOR/GORD before could know the 'right' things to say to feel heard and understood, or alternatively, health professionals could be more trusting. This area warrants further exploration.

Existing literature describes that continuity of care can be challenging and multi-layered, especially for neonates transitioning home from intensive care units [37, 38]. This was evident in parents of preterm infants, where participants felt that they were dismissed by other health professionals such as General Practitioners if they were under the continued care of paediatric or neonatal teams. A New Zealand study found that having a designated case manager could aid seamless multi-disciplinary care [38], which could be considered in the UK.

Participants described difficulties with validity and reliability of information on GOR/GORD available on the internet, and particularly the challenges in appraising the quality of evidence. This is a well-recognised and long standing issue for the general public accessing medical information [39]. However, it is perhaps even more pertinent for infant conditions such as GOR/GORD, where management is not clear cut and many parents seek out possible solutions to improve signs and symptoms.

The use of qualitative methodology is a key strength of this study, allowing in depth exploration of a topic with limited research in this setting and where surveys have previously dominated. Triangulation through double coding of transcripts enhances the credibility of the study and provides assurance that thematic analysis represents participants' voices [26]. However, several limitations are also acknowledged. The sample size is smaller than planned as, despite pursuing several avenues of recruitment, responses were limited. Convenience sampling was used which limited the breadth of demographics of those interviewed, however qualitative methods do not seek to achieve representative sampling or generalisable findings. Furthermore, such research is novel particularly within the UK setting and our findings highlight areas for improvement from both parent and healthcare professional perspectives (e.g. more consistent information needed for the former and training to address varying knowledge gaps for the latter). Whilst data saturation was reached for the most part, only one father was interviewed, and the study appeared to attract more parents of children with GORD rather than GOR. As a result of these limitations, study findings might lack transferability to other individuals or settings [26].

## Conclusions

In conclusion, infant GOR/GORD can affect infants and parents in a variety of ways, impacting both physical and mental health. GOR/GORD is complex to manage, with difficulties administering medications that have varied effectiveness, and symptoms that overlap 'normal' infant behaviour. Parents play a vital role in management, but there is lack of consistency of information, and levels of knowledge about GOR/GORD among healthcare professionals vary. More education on the diagnosis and management of reflux could be beneficial for parents-to-be, parents and clinicians, and further research is needed into healthcare professionals' knowledge and perceptions.

More teaching could be introduced to the training curriculum of healthcare staff, including awareness of the parental impact and the importance of managing expectations. Additionally, it could be beneficial to develop a clearer definition of GOR and its distinction from GORD, with clear evidence-based diagnostic criteria and management guidance.

Qualitative studies with fathers, to explicitly explore their experiences of infant reflux, and with health professionals (including General Practitioners, health visitors and pharmacists) to explore their knowledge and perceptions of infant GOR would help to increase understanding.

## Supporting information

**S1 File. Semi-structured interview topic guide.**
(DOCX)

## Author Contributions

**Conceptualization:** Kathryn McVicar, Lisa Szatkowski, Shalini Ojha, Manpreet Bains.

**Data curation:** Kathryn McVicar, Lisa Szatkowski, Shalini Ojha, Manpreet Bains.

**Formal analysis:** Kathryn McVicar, Lisa Szatkowski, Simon Tunster, Manpreet Bains.

**Investigation:** Kathryn McVicar, Shalini Ojha.

**Methodology:** Kathryn McVicar, Lisa Szatkowski, Shalini Ojha, Manpreet Bains.

**Project administration:** Kathryn McVicar, Lisa Szatkowski.

**Software:** Kathryn McVicar.

**Supervision:** Lisa Szatkowski, Shalini Ojha, Manpreet Bains.

**Validation:** Lisa Szatkowski, Simon Tunster, Manpreet Bains.

**Visualization:** Kathryn McVicar.

**Writing – original draft:** Kathryn McVicar.

**Writing – review & editing:** Kathryn McVicar, Lisa Szatkowski, Shalini Ojha, Simon Tunster, Manpreet Bains.

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
