## [Decision Letter · Decision Letter 0]

24 May 2024

PONE-D-24-01188Exploring parents’ experiences, attitudes and understanding of gastro-oesophageal reflux in infantsPLOS ONE

Dear Dr. Bains,

Thank you for submitting your manuscript to PLOS ONE. After careful consideration, we feel that it has merit but does not fully meet PLOS ONE’s publication criteria as it currently stands. Therefore, we invite you to submit a revised version of the manuscript that addresses the points raised during the review process.

We look forward to receiving your revised manuscript.

Kind regards,

Jerry Zhou, Ph.D.

Academic Editor

PLOS ONE

2. In the online submission form, you indicated that [The data underlying the results presented in the study are available from the authors, upon request, for researchers who meet the criteria for access to confidential data.]. 

Reviewers' comments:

Reviewer's Responses to Questions

**Comments to the Author**

1. Is the manuscript technically sound, and do the data support the conclusions?

Reviewer #1: Partly

Reviewer #2: Partly

2. Has the statistical analysis been performed appropriately and rigorously? 

Reviewer #1: No

Reviewer #2: N/A

3. Have the authors made all data underlying the findings in their manuscript fully available?

Reviewer #1: Yes

Reviewer #2: Yes

4. Is the manuscript presented in an intelligible fashion and written in standard English?

Reviewer #1: Yes

Reviewer #2: Yes

5. Review Comments to the Author

Reviewer #1: The authors conducted a study to determine parental experiences, attitudes and understanding of the symptoms,

diagnosis and management of infant GOR. There are several concerns that needs to be addressed.

1.The population may not be representative as convenience sampling was used

2.The sample size of the study is very small, hence the conclusion drawn is not significant.

3.Please justify the research question and how this study adds to the current literature

Reviewer #2: Overall I feel the authors have delivered on their goal in providing a snippet into the experiences and attitudes of parents with children with GORD. The study is somewhat limited by the small numbers though this is acknowledged by the authors.

6. PLOS authors have the option to publish the peer review history of their article (what does this mean?). If published, this will include your full peer review and any attached files.

Reviewer #1: **Yes: **Jeyasakthy Saniasiaya

Reviewer #2: No

---

## [Author Response · Author response to Decision Letter 0]

25 Jul 2024

We have attached a detailed response to reviewer comments with our revised document. Have pasted here:

Reviewer WW comments sent via PDF

Reviewer comment: pg 3, L44/45 can say lower oesophageal sphincter (LES)?

Authors’ response: This has been amended as suggested.

Reviewer comment: pg3 L48/49 whilst?

Authors’ response: Amended, as suggested.

Reviewer comment: Pg 4 L70 what does this statement mean?

Authors’ response: This statement has been edited to read ‘A number of studies have shown the importance of parents in both diagnosis and management[5,6,13].’

Reviewer comment: pg 4 L70 dont think parenthesis are needed here (referring to L69).

Authors’ response: Parenthesis removed.

Reviewer comment: pg 5 L113-115 i think this point is a bit redundant. seems presumed.

Authors’ response: Statement has been removed.

Reviewer comment: p6 L126/127 i think maybe re-word this a bit? the sentence makes it seem like you didn't read the transcripts properly in the first instance.

Authors’ response: The statement now reads as, ‘Firstly, transcripts were read and re-read to develop familiarity, and initial reflections were documented.’ 

Reviewer comment: p9 L198 is this specific to reflux though

Authors’ response: Parents did link expectations not matching up to expectations due to symptoms of reflux. The sentence has been amended to read as, ‘Parents explained the impact that infant GOR/GORD had had on their parenting experiences, and how consequently this had not matched up to their expectations, as parents explained that the symptoms of reflux had affected their bonding with their child.’

Reviewer comment: p14 L315 what proportion/how many parents expressed sentiments with the themes mentioned above?

Authors’ response: It is not normal practice to report frequencies for data analysed in a purely qualitative manner, and thus language is better suited to provide insights. Having said this, qualitative research is not necessarily concerned with proportions, instead we strive to reflect the breadth and depth of accounts. We analysed data using Braun and Clarke’s approach for thematic analysis, rather than a summative content analysis (which is an approach that combines both quantitative and qualitative components). Finally, given the small sample size, proportions can be misleading and/or allow for disclosure of who said what in the case where just one or two people gave a particular response.

Reviewer comment: p14 L318 what number/proportion? would be better to have objectivity rather than 'many'

Authors’ response: Above response applies here. We have however, amended text in places to use language to better reflect views of group as a whole e.g. here we have changed ‘many’ to ‘most’.

Reviewer comment: p18 L421 i would be careful with this statement as it fairly inflammatory

Authors’ response: The sentence has been amended to read, ‘Parents who have experienced GOR/GORD before could know the ‘right’ things to say to feel heard and understood, or alternatively, health professionals could be more trusting.’

Additional comments sent in body of e-mail:

Reviewer comments: The population may not be representative as convenience sampling was used. The sample size of the study is very small, hence the conclusion drawn is not significant.

Authors’ response: The two points above have been addressed to highlight that despite limitations with sampling approach and sample size (addressed as limitations), our findings do offer something novel in the area. The following has been added to the discussion section:

‘The sample size is smaller than planned as, despite pursuing several avenues of recruitment, responses were limited. ‘Convenience sampling was used which limited the breadth of demographics of those interviewed, however qualitative methods do not seek to achieve representative sampling or generalisable findings. Furthermore, such research is novel particularly within the UK setting and our findings highlight areas for improvement from both parent and healthcare professional perspectives (e.g. more consistent information needed for the former and training to address varying knowledge gaps for the latter).’ 

Reviewer comment: Please justify the research question and how this study adds to the current literature.

Authors’ response: We have added to the Introduction section to highlight where current gaps are, to help justify the research question. ‘A number of studies have shown the importance of parents in both diagnosis and management[5,6,13], but research into parental knowledge and experiences of GOR/GORD is limited (e.g. in the United Kingdom).’ 

And again, ‘Given the recognised importance of parents in the diagnosis and management of GOR/GORD, and lack of exploration into their attitudes and experiences, further research is needed[5,6,13], to understand whether findings from other settings are transferable to the UK, and to identify where gaps in knowledge are and ways to better support parents in the diagnosis and management of GOR/GORD (including who could input into this and how). The purpose of this study is to explore parental experiences, attitudes and understanding of the symptoms, diagnosis and management of infant GOR and GORD. This study is needed to begin to identify ways forward for future approaches to parental education and support, which would faciliate GOR/GORD management[17].’ 

Changes to the Discussion e.g. including the above adds commentary, or that we have rephrased prior sections to make more explicit what this work adds to the literature:

‘A novel finding from our study is that GOR/GORD can impact a mother’s decision to breast feed future children, an additional factor that could affect parental mental wellbeing[33].’

‘Crucially, our work suggests a lack of consistency in the knowledge and attitudes among healthcare professionals regarding infant GOR/GORD.’

‘Another novel finding from our work was the marked difference in attitudes from professionals between first-born and subsequent children which has not been previously identified in this context. This could reflect the increased parental knowledge and experience of having a previous child with GOR/GORD, and links with the parent-driven nature of management.’

---

## [Decision Letter · Decision Letter 1]

6 Aug 2024

Exploring parents’ experiences, attitudes and understanding of gastro-oesophageal reflux in infants

PONE-D-24-01188R1

Dear Dr. Bains,

We’re pleased to inform you that your manuscript has been judged scientifically suitable for publication and will be formally accepted for publication once it meets all outstanding technical requirements.

Kind regards,

Jerry Zhou, Ph.D.

Academic Editor

PLOS ONE

Additional Editor Comments (optional):

Reviewers' comments:

Reviewer's Responses to Questions

**Comments to the Author**

1. If the authors have adequately addressed your comments raised in a previous round of review and you feel that this manuscript is now acceptable for publication, you may indicate that here to bypass the “Comments to the Author” section, enter your conflict of interest statement in the “Confidential to Editor” section, and submit your "Accept" recommendation.

Reviewer #1: All comments have been addressed

2. Is the manuscript technically sound, and do the data support the conclusions?

Reviewer #1: Yes

3. Has the statistical analysis been performed appropriately and rigorously? 

Reviewer #1: N/A

4. Have the authors made all data underlying the findings in their manuscript fully available?

Reviewer #1: Yes

5. Is the manuscript presented in an intelligible fashion and written in standard English?

Reviewer #1: Yes

6. Review Comments to the Author

Reviewer #1: the authors present an interesting an important study.

The authors have addressed all the comments adequately.

7. PLOS authors have the option to publish the peer review history of their article (what does this mean?). If published, this will include your full peer review and any attached files.

Reviewer #1: **Yes: **Jeyasakthy Saniasiaya

---

## [Editor Report · Acceptance letter]

12 Sep 2024

PONE-D-24-01188R1 

PLOS ONE

Dear Dr. Bains, 

I'm pleased to inform you that your manuscript has been deemed suitable for publication in PLOS ONE. Congratulations! Your manuscript is now being handed over to our production team.

Kind regards, 

on behalf of

Dr. Jerry Zhou 

Academic Editor

PLOS ONE